# Experience of Indigenous Peoples’ Access to Long-Term Care Services in Taiwan: A Qualitative Study among Bunun Tribes

**DOI:** 10.3390/healthcare10122383

**Published:** 2022-11-27

**Authors:** Hsiu-Chuan Tien, Wen-Li Hou, Yung-Mei Yang

**Affiliations:** 1School of Nursing, College of Nursing, Kaohsiung Medical University, Kaohsiung 80708, Taiwan; 2Department of Senior Citizen Services, National Tainan Junior College of Nursing, Tainan 700007, Taiwan; 3Department of Medical Research, Kaohsiung Medical University Hospital, Kaohsiung 80708, Taiwan

**Keywords:** Taiwanese indigenous people, indigenous tribe, long-term care, culturally appropriate care

## Abstract

Indigenous communities usually have poorer access to long-term care services than non-indigenous communities because of their remote locations and unique cultural backgrounds. However, there was little exploration into the experience of indigenous people’s access to the official long-term care services in Taiwan—the gap this study aimed to fill. A qualitative study design using semi-structured interviews was used to obtain data from a purposive sample. Fourteen participants who were disabled and lived among the indigenous communities of the Bunun tribes in central Taiwan were interviewed individually. The data were analyzed using Graneheim and Lundman’s qualitative content analysis. The theme—“helpful but still difficult and unfit”—and three categories with eight subcategories emerged. While official long-term care services provided by the government can benefit people with disabilities in indigenous tribes, their use of such services faces a number of obstacles, which points to the need for considering culturally appropriate care. To protect the rights and interests of indigenous tribal communities, long-term care policies and practical planning must be adopted, cultural differences at play must be respected and recognized, and the necessary support must be offered to eliminate inequalities in healthcare.

## 1. Introduction

In several nations, disabilities are approximately 2–10% more common among the indigenous population than the general population [1,2,3,4]. Because indigenous people have unique cultural, economic, and social structures, their healthcare requires more support. However, indigenous people often experience obstacles and inequality in terms of the acquisition of healthcare services [5,6,7]. In Taiwan, the aging index—that is, as a ratio between those aged 65 and over and those aged 0 to 14—has increased year over year among indigenous people, growing from 26.7% in 2009 to 41.9% in 2019 [8]. Similarly, the number of people with disabilities in Taiwan has also been estimated to increase every year. As a result, the demand for long-term care for indigenous people in Taiwan is likely to increase.

As the fastest-growing aging population in the world, Taiwan entered the status of an aged society in 2018 and is projected to grow to a super-aged society—a society having 20% of its population in the 65-or-over group—in 2025 [9]. In light of the soaring demand for long-term care, the Taiwanese government initiated its first National Ten-Year Long-Term Care Plan 1.0 (LTC 1.0) in 2007, which laid the groundwork for Taiwan’s system and model for care services. The Long-Term Care Services Act was also promulgated in 2015 to protect the development of the system of care services through government legislation. When the LTC 1.0 was adopted, the Taiwanese government realized that the policy required some amendments to properly address public needs. To cater to the care needs of different populations, a service user-centric, diverse, universal, and continuous system of care services was established. In 2016, the Ten-Year Long-Term Care Plan 2.0 (LTC 2.0) was passed in Taiwan, which classified long-term care into care and professional services, the transportation services, assistive devices and improvement of the domestic barrier-free environment, and respite-care services. Compared to the LTC 1.0, the LTC 2.0 featured an expanded scope of service targets and items, and more flexible and selective services. The construction of the comprehensive community-based care model also enhanced the accessibility and continuity of services. In terms of service charges, low-income households are fully borne by the government, and the others need to bear part of the shared expenses by themselves, which is approximately 5–30% of care service fees [10].

However, limitations in environmental and social structures, such as a shortages of service resources, inconvenient transportation services [11,12], and financial vulnerabilities [11], pose challenges to the accessibility of care services in indigenous villages. Moreover, formal resource allocations are often outsourced, and care service operations are more difficult to carry out in indigenous villages than in local organizations [13]. Notably, owing to the linguistic and cultural gaps between care service providers and indigenous communities, services often fail to meet the needs of cases and their families in indigenous villages [14,15]. Accordingly, the LTC 2.0 established a “special chapter for long-term care of indigenous peoples” and invited indigenous groups and experts to discuss obstacles and establish plans for the development of care services for indigenous tribes [16]. Ultimately, it aims to find solutions to the obstacles by providing effective services in indigenous communities and ensuring culturally appropriate care policies and practice. However, it remains unclear whether Taiwan has solved these issues related to care among indigenous communities since the LTC 2.0.

Taiwan’s indigenous populations are Austronesian people who are, according to official statistics, classified into 16 ethnic groups according to their unique cultural customs. There are about 569,000 indigenous people accounting for 2.38% of the total population in the country. Having dispersed across both sides of the Central Mountain Range, the Bunun people are the fourth largest indigenous group in the Taiwanese population. With their settlements located in remote mountainous areas, they also reside at the highest elevations among ethnic groups in Southeast Asia. Their traditional cultural norms and taboos remain inseparable from the lives and well-being of tribal members to this date. Most Bunun tribes make a living by farming and consider “sharing” and “mutual help” to be important social and cultural values. For instance, they jointly work to meet labor demands during the busy farming season; meanwhile, they also establish a mutual savings bank to collect capital to purchase the equipment required to cultivate crops [17]. Notably, the Bunun people conform to a strict patriarchal clan system called *sidoq*. The clan shares hunting grounds, food, and work; observes funeral rules; and respects clan elders. Historically, larger families have typically shared the responsibility of caring for their elders [17]. However, young adults are increasingly moving from indigenous villages to urban areas for work; this has caused the caregiving function of the family [14,18], and the comprehensive care function of tribes to wane over time. As of August 2022, there were approximately 16 local long-term care service stations among the Bunun tribes in central Taiwan; notably, these communities most frequently use care services, followed by transportation services [19,20].

Despite the implementation of the LTC 2.0, the use of care services among indigenous tribes has remained under-studied in Taiwan; the status of their use of these services urgently needs to be addressed. Thus, this study explored indigenous villagers’ experiences with using long-term care services in Taiwan and their care needs to clarify the situation of care in Taiwan’s indigenous villages. Specifically, we collected insights from service users in a Bunun tribe living in a remote mountainous area in Taiwan.

## 2. Materials and Methods

A qualitative study was conducted to explore the experiences of Taiwan’s indigenous tribal communities regarding their access to and use of public long-term care services.

### 2.1. Participants and Recruitment

Research participants were selected from an indigenous village of a Bunun tribe in central Taiwan. Interviews were conducted from December 2019 to July 2021. Participants were required to be members of the Bunun tribe and have used at least one public long-term care service within the past three years. Individuals were excluded if they were under 20 years of age or had a cognitive, expressive, or mental disorder.

With the help of local case managers, tribal elders, pastors, and care service providers, the first researcher explained the background and purpose of this study to each participant who met the inclusion criteria and invited them to participate. They were not required to respond or give consent immediately during the first round of invitations; they were advised that the researcher would also pay them a second, unaccompanied visit to confirm their willingness to participate. Alternatively, they could choose to initiate a response indicating their decision. Those who agreed to participate in this study were contacted individually to determine the location, date, and time of the interview as per their preferences.

### 2.2. Ethical Considerations

This study was approved by the Human Research Ethics of National Cheng Kung University (Approval No. NCKU HERC-E-108-349-2). The interviews were conducted after the participants were informed of the details of this study, agreed to participate, and signed a consent form. The participants could withdraw from the research at any time. Further, this study was made confidential, and those who were not associated with the research had no access to the interview data.

### 2.3. Data Collection

This study adopted purposive sampling and snowball sampling. The researcher explained the purposes of the study to qualified participants. Upon their agreement, one-on-one semi-structured interviews were conducted. Each interview lasted for 60 to 90 min and was recorded. Data collection continued until the content was saturated. The interview questions focused on the demand for long-term care service needs, use, and difficulties.

### 2.4. Data Analysis

Collected data were analyzed using the *qualitative content analysis* method *proposed by* Graneheim and Lundman. The procedure consisted of the following steps: (1) interviewees read the text several times to understand the overall context; (2) meaning units in the participants’ experiences with long-term care services were established; (3) meaning units were condensed while preserving the core meanings; (4) condensed meaning units were labeled with an initial code and classified into sub-categories; and (5) sub-categories were abstracted into categories, which eventually led to the formulation of a theme.

### 2.5. Rigor and Trustworthiness

We examined the rigor and trustworthiness of this study based on the four criteria proposed by Guba and Lincoln on the precision of qualitative research, as follows: (1) Credibility: the first researcher was a local Bunun person with experience of working in long-term care services among indigenous tribes, familiarity with the Bunun cultural customs, and the ability to easily establish a trusting relationship with the subjects. Further, the researcher received training in qualitative research with the experience of practicing interviews and qualitative analyses. Throughout the research process, the researcher regularly held discussions with experts in qualitative research. (2) Transferability: indigenous users of care services in indigenous communities described their perceptions of and experiences with the services; their descriptions offered meaningful insights into the situation of care services among indigenous tribes in Taiwan, which can serve as a reference for future care practices. (3) Dependability: multiple rounds of testing were conducted for textual interpretations, meaning units, coding, and subcategory and category labelling to yield themes until the interviewer reached a consensus with two experts in qualitative research on the research findings. (4) Conformability: when the subjects’ descriptions were ambiguous, the researcher immediately clarified them with the participants. Finally, the participants reviewed and confirmed the research findings.

## 3. Results

A total of 14 participants were enrolled, consisting of 9 males and 5 females aged 43–77 years who had used services for 1 to 10 years, including home services, professional-care services, and transportation services (Table 1).

According to the research findings, one theme—“Helpful but still difficult and unfit”—and three categories with eight subcategories were identified (Table 2). Being helpful but still difficult and unfit was the core element across users’ experiences; this suggests that care services do help indigenous people in Taiwan, but still involve issues in terms of accessibility, availability, affordability, accommodation, and acceptability.

### 3.1. Barriers to Accessing Long-Term Care Services

The participants experienced barriers to accessing services, including a lack of understanding about service information, insufficient service resources, restrictions on service items, and economic difficulties.

#### 3.1.1. Lack of Understanding about Service Information

Participants were generally unfamiliar with information on public long-term care services and were often uncertain about what the service items entailed or how to apply for such services. Two participants raised the following doubts about applying for respite services and assistive devices:


*There are many service items in long-term care. I am using home services currently. I remember that there is something called a “respite care service.” Which can help family members. I have heard of it, but I’ve never used this service and don’t know whether I can apply for it or how to apply for it.*
(P1)


*I cannot move easily…I’m thinking if I had a mobility scooter, I would go out for a ride instead of staying home all day…However, I have no idea how to apply for one or who I should ask for help.*
(P9)

Owing to a lack of understanding about long-term care service, the majority of participants were not aware of certain services until local service providers reached out and explained them. In this step, these participants only accepted the services passively after being asked about their willingness to use them.


*After I lost the ability to walk, they (Bunun service providers) came to me and told me about a subsidized long-term care scheme that could look after me…Since then, I’ve been in their care for more than a year now. Before they came, I had no idea there was a government-funded service that could help me.*
(P4)


*At the time, my husband and I had physical disabilities, so they reached out and came here to ask if we wanted to use their services. My children later agreed, and they started coming here to perform the services.*
(P13)

#### 3.1.2. Insufficient Service Resources

Owing to the shortage of workers and material resources, the overall service capacity in indigenous villages has been falling short of demand. The shortage of services was particularly pronounced in home care and transportation services.

Participants used a home care service for one-to-two hours twice a week. They asked for longer service hours, but service units were unable to satisfy this request due to staffing shortages. Sometimes, they even adjusted service providers out of consideration for their limited staff. Although they were frustrated by the situation, the participants had no choice but to compromise.


*I prefer more service hours, which will gives my wife plenty of time to go out and attend to family matters…but there’s currently a huge shortage of home caregivers…More effort should be put into promotions and getting more people into this line of work. This would give service units more options in allocating staff, service hours, and services…The problem right now is that labor supply is rather tight!*
(P1)


*Today, my home caregiver told me that this is her last day. She said her supervisor had transferred her to a different position because of the shortage of helping hands…It’s sad to see her go, but since the arrangement has been made by her supervisor, I can only cooperate…This had happened more than once. They’ve changed my home caregiver two or three times already…*
(P10)

Due to the limited availability of vehicles for transportation services, participants had to make a reservation with the driver and followed the vehicle dispatch and routing arrangements of service units. Participants lived in a remote mountainous area and had to travel a long way for medical appointments. Additionally, following the routing arrangements of the dispatching units prolonged travel time and often left participants physically exhausted.


*I have to return for an appointment every month, and, before each appointment, I have to tell him (the service provider) which date and month it would be. We have to arrange our medical appointments based on his availability to dispatch a vehicle and must avoid scheduling it during his off-duty hours or meetings. A lot of people need a ride…Sometimes, I get in the car first. However, since he has to pick up other patients along the way, I am often the last one to arrive at the hospital…After a long journey sitting in the car, I often come home completely shattered…*
(P5)

Occasionally, service users have to reschedule their home services based on drivers’ availability. This can leave service users to feel helpless and indirectly be a burden on the home care service unit.


*Sometimes, to work around the vehicle service schedule, we are forced to rearrange our home caregivers shifts. This is the hassle of hailing a vehicle…We feel awfully sorry for causing them trouble, but what else can we do? This is our only option when there aren’t enough cars.*
(P4)

Conversely, owing to the scarcity of resources, service users in need of last-minute medical consultations usually struggled to book an available vehicle.


*I was planning to go to the hospital yesterday, but I couldn’t get a car no matter who I called—the county government or the district office. The reservations were all full to the brim! It was such a headache…*
(P4)


*Since I am in a wheelchair, a car with a lift is more convenient. However, I must make a reservation in advance…It’s usually impossible to book a car if I feel unwell and have to make a last-minute doctor’s appointment…*
(P7)

Lastly, in addition to the limited availability of serviceable vehicles, there were also limitations in the medical facilities they could provide. As a result, service users were often left without much choice and had to compromise and settle for medical facilities where the government transportation service operated.


*Originally, I wanted to go to another hospital, but the transportation service doesn’t operate there. So I had to come to this hospital where the service operates …*
(P2)

#### 3.1.3. Restrictions on Service Items

Public care services of all types were bound by standardized regulations that hindered the participants’ use of services. For example, while home caregivers provided general everyday care, they did not change wound dressings. Even if they wanted to help service users change wound dressings that had been soaked after an assisted shower, they had to comply with regulations and resort to asking the service users to change the dressing. This led to the following event: Participant 12 had wounds from gouty arthritis in his arms and legs. Although a nurse had been scheduled for weekly home visits and wound care guidance, the participant was living with his elderly mother who was powerless to help. As local medical resources were not as accessible as those in urban areas, the participant often had to wait for the nurse’s weekly visit to have his wounds redressed.


*My main concern right now is my wounds. Flies are rampant in summer, and I’m worried there will be a problem if I don’t change my wound dressings…I have no way of doing it myself, and my mother is too old to help me do it. There’s no clinic nearby, and it’s such a hassle to go out and seek medical attention. I thought home caregivers could do me a favor by helping me take a shower, but they’re prohibited from helping anyone change wound dressings…What else can I do? It is what it is…*
(P12)

#### 3.1.4. Economic Difficulties

Apart from the shortage of transportation service resources, limited financial resources were the most common obstacle to accessing services. Most locals in the study worked in farming and labor. In the event of a loss of income, they had to rely on social welfare or family members. Thus, they often had trouble purchasing the services they required.


*They help me take a shower and clean the house. It costs around NTD 1000 per month…One of my legs was injured and I could not work by climbing the mountain anymore…My entire livelihood depends on the meager disability allowance I receive…Even though I want more services, I have no way to affording them.*
(P6)


*I used to carry metal for a living. Since I had open surgery for my lower back injury, I haven’t been able to walk or work…Now, I have to scrape a living by claiming benefits from the old-age guaranteed pension payment. Sometimes, I have to give up going to the doctor’s because I dread having to pay for the carfare!*
(P5)


*According to the service contract, they give me a sponge bath. I hope that they can assist my wife with the housework. Moreover, I also hope that they can increase my rehabilitation treatments. However, this will cost more money, which would burden my wife.*
(P4)

### 3.2. The Need for Culturally Appropriate Long-Term Care Services

The participants expressed their need for culturally appropriate long-term care, including a preference for service providers from the same ethnic group and a desire for mental and spiritual comfort.

#### 3.2.1. Preference for Service Providers from the Same Ethnic Group

The participants preferred service providers from the same ethnic group mainly because, as they put it, they felt more comfortable with someone who shared their language and style of communication. They reported having difficulties cultivating meaningful relationships and interactions with non-indigenous providers.


*It doesn’t matter whether the service is from the Han or the Bunun people. However, I prefer the Bunun as it is easier to get along with them. On the contrary, Han people always seem prejudiced.*
(P8)


*I prefer the Bunun. We speak the same language and can chat with each other…There was an outsider who came here on the weekends. He usually left after working for one hour. Because of the language barrier, we didn’t know what to say to each other…*
(P3)


*I made jokes with the Bunun service providers, and we could easily get along with each other since we share the same culture. Interacting with the Han service providers is less relaxing.*
(P1)


*When I made a return visit to the hospital in his car for the first time, I thought he was a Han and I did not talk to him. When we arrived at the hospital, he helped me get out of the car and he reminded me to be careful in the Bunun language. I realized that he was a Bunun, too. Afterward, when I was in his car, I would chat with him.*
(P5)

The participants gave many accounts of their relationships and interactions with Bunun service providers outside their rigid contractual relationships, which were characterized by mutual empathy, acceptance, care, accommodation, and consideration. They considered service providers from the same tribe or ethnic group as “insiders” and were grateful for and compassionate toward their hard work. The service providers were also attentive to their care needs and actively offered them assistance both within and outside the contracted scope of services.


*She (a Bunun home service provider) knows my habits like an insider. She also treats me like a family member and would exchange pleasantries with me and ask about my life while she is working. She doesn’t just see it as a job and leave when she’s done. Sometimes, I would call and ask her to come earlier the next day because of personal matters. Even though it is earlier than her working hours, she’s willing to accommodate my needs…Sometimes, riding a motorcycle in the mountainous area can get very dangerous when it’s raining, and she has to report to her next case after finishing her job here. I’d tell her to leave 10 min early even though the service session isn’t over because I’m worried that she might slip and fall on her way…*
(P1)


*They’re [Bunun home service providers] all very kind…When it’s really cold in winter, she will take my quilt outside to the sunlight. She said, this way, the quilt can keep me warm when I put it on. This isn’t actually part of the contract…We’re accommodating each other’s needs. Sometimes, if she arrives an hour late because she hasn’t finished her last case, she will call and tell me in advance and I will tell her to take it easy and take her time…*
(P5)

Similar scenarios were reported regarding the use of transportation services for medical appointments: participants recalled how Bunun service providers accommodated their needs with flexibility whenever necessary. For instance, participants reported that Bunun transportation service providers would wait for them if their appointments ran late—even if this meant staying past service hours without overtime pay. Given the long travel necessary for medical appointments, this gave the participants peace of mind.


*The Bunun driver is more easygoing and friendly. He doesn’t just leave as soon as his time is up. He’d wait until our appointment is over knowing full well that he would be working overtime without pay. Sometimes if we want to buy something on our way, he’d kindly pull over and wait until we’re finished.*
(P14)


*I still prefer our own Bunun driver. He’s more understanding and doesn’t insist on his own interests. He would wait until our appointment is over and doesn’t stop work the moment his time is up, which was reassuring for us. Sometimes, he’d painstakingly wait until it’s very late, so we would buy him a takeaway meal.*
(P5)

Furthermore, service providers from the same ethnic group shared the same lifestyle, background, and culture with the participants and understood their preferences and daily habits. Thus, they could provide culturally appropriate care services that gave the participants a sense of happiness in their life. One of the participants, for example, listed some of his favorite traditional indigenous dishes that were prepared by the Bunun carer:


*I like the “Grandma’s Soup” prepared by Salung (Bunun home service provider’s name). It’s basically pigeon pea, a type of bean used in indigenous cooking, slow-cooked with wild vegetables and ginger in a pot. It smells really good. I just call it Grandma’s Soup. When you use game meat, it’s called “Grandpa’s Soup.” Just add a bit of tana, and the smell is so inviting that you can devour two more bowls of rice with it…And then there are the grilled salted fish and salt-cured pork…Ah, they were so good! I was so blessed.*
(P2)

#### 3.2.2. The Desire for Mental and Spiritual Comfort

Living with a disability often put the participants in situations that caused them physical and mental suffering. Therefore, they longed for and sought refuge in religious beliefs, from which they gained the power to persevere and a sense of spiritual comfort.


*Sometimes, it feels like life is getting overwhelming and so frustrating that I don’t know what to do. Whenever I’m in agony and on the verge of giving up, I pray. A few days later, someone would come to my aid. That’s why I’m very grateful and convinced that God exists.*
(P5)


*After I had an amputation, I had a hard time coming to terms with it and became suicidal. I’d shut myself in my home and refuse to go out. Eventually, praying was how I slowly found my way out…I kept thinking to myself: I’d be long gone if it weren’t for my belief in God.*
(P9)


*My leg has been in a lot of pain over the last year. It tingles so much that I can’t walk. But the doctor said an open surgery would be dangerous. I was truly terrified, so I prayed as hard as I could and asked God to show me the way to inner peace.*
(P11)

Bunun care service providers understood the spiritual demands of the participants. When they comforted the participants by reciting poetry, prayers, and Bible verses during the care services, the participants moods improved.


*Sometimes, she [ Bunun home service provider] would sing and pray with us, and through the singing and prayers, I felt my heart calm down.*
(P2)


*She chatted with me while working…Sometimes, when I was talking about the difficulties in my life, she could feel my grief and would encourage me by reciting the Bible verses. To be honest, I was comforted by that.*
(P5)

In addition, tribal churches also offered assistance and care in various forms (e.g., physical materials, spiritual consolation). Thus, they also played an indispensable role for the participants.


*There are two churches in our tribe. Members often come to offer us their regards and prayers. During certain festivals, they would even give us each a red packet. The point isn’t the money inside, but the blessing it represents. It matters a lot to us, as it tells us that the tribe still knows about our existence. If this type of care didn’t exist, I, as a person, would have ceased to exist as well.*
(P1)


*The churches would help us and come here to offer prayers. They bring me great peace of mind and give me joy that there’s still someone who cares for us.*
(P5)

### 3.3. Benefits of Long-Term Care Services

Overall, the participants were grateful and appreciative of the benefits that long-term care services had brought to their lives. The perceived benefits from such services included physical and mental comfort and reduced caregiver burden.

#### 3.3.1. Physical and Mental Comfort

Many participants reported a sense of physical and mental comfort as a transformation brought about by the services. By offering assistance with their everyday lives, the long-term care services released them from their suffering. One of the users of the home services said the following:


*Before this service came into my life, I would be in a soiled condition for hours on end without someone changing my diaper for me. I was hungry and thirsty…My wife was the breadwinner and couldn’t always tend to me. Then, this government service came along to help. People bathed me and sent me ready-made meals. It has eased my mind and kept me from thinking that my condition is going downhill…I do feel more reassured, to be honest! If it weren’t for this, our family would probably be like the people you hear about in the news…they would snap under the stress of long-term caregiving and push us in our wheelchairs into the ditches…*
(P4)

According to a blind participant with mobility issues who was living alone:


*Back then, I’d always struggled to find my way out, but my heart was settled once I came here (the tribal foster care unit). People encouraged me to be optimistic and were always been incredibly helpful. They helped me shower, took care of my clothes and shoes, prepared meals for me, and even laid down the tableware neatly in front of me…When I was sick, they escorted me to the hospital. They got me whatever I needed. My life is so much better here. I am no longer lonely …I have started going back to my jolly old self…*
(P2)

#### 3.3.2. Reduced Caregiver Burden

The families of people with disabilities lived in an atmosphere of suffocating stress, which placed a substantial physical and mental strain on both the participants and their family caregivers. As a result, verbal exchanges were often accompanied by outpourings of negative emotions.


*When my wife and children felt exhausted from caring for me, they would murmur: “Why don’t you just die? If you have already passed away, we wouldn’t have any sickness in the family. Then, we would be free from stress.”*
(P4)

By offering help with personal hygiene, homemaking assistance, and companionship, services temporarily relieve family caregivers of the responsibilities of caregiving, enabling them to run household errands and go to work with their minds at ease. Overall, the services reduced the burden on family caregivers of people with disabilities.


*The largest benefit has been the reduced burden on my wife as a caregiver…She can save the time she would spent caring for me and use it to sort out other things at home.*
(P1)


*They’ve helped us a lot by coming here to serve us. My wife is under less stress from caring for me and can go to work with less worry.*
(P4)

## 4. Discussion

This study found that the members of the indigenous tribal community in this study face many obstacles in using long-term care services in Taiwan. First, participants demonstrated limited understandings of services, which is consistent with the finding of Hou and Kuo [15]. This lack of related information has influenced their ability to apply for and use services. Among indigenous tribes in Taiwan, information is mainly disseminated through interpersonal communications and exchanges. Using newspapers or other forms of mass media as the dominant mode of information dissemination often leads to difficulties in receiving and understanding the information conveyed. Therefore, we suggest that information about services should be disseminated in the way the tribe members are used to exchanging information, such as through church gatherings, clan gatherings, and intra-tribal festival gatherings for dissemination. Moreover, local long-term care managers also need to actively identify the potential service users using face-to-face explanations. Second, the shortage of care resources in these tribes had led to circumstances in which demand was not necessarily met by service availability; these circumstances resembled the picture of indigenous villages painted by Franco et al. [21] and Lin and Huang [22]. Third, even when care services are within their arm’s reach, older adults and people with physical and mental disabilities within indigenous communities in Taiwan may struggle to afford them as a result of their limited incomes and poorer economic conditions. Lastly, even if they can access these services, they may encounter difficulties in using them due to problems related to services because of the problems with service appropriateness; this is, again, in line with the finding of Hou and Kuo [15].

Cultural safety is an indispensable concern in indigenous care. Service providers with the same cultural background were more reliable for the participants and could break down the barrier established by the service contract. They were able to provide culturally appropriate care by showing compassion, knowing and respecting the people they served, and considering their cultural needs when formulating care plans. Contrarily, constructing relations with individuals when service providers are outsiders with different languages and cultural backgrounds is difficult. This finding is consistent with the research findings of Larke et al. and Topp et al. [23,24]. This phenomenon shows that the reinforcement of culturally appropriate care improved accommodation to care [25], and that accordingly upgrading care services is also an important step in the direction of maintaining the quality of indigenous villages [26]. Through solid training, first-line service providers can learn to recognize and respect cultural differences and provide culturally appropriate care [27].

Moreover, the tribe resembles a big family, where people share a close relationship. The cultural background of mutual sharing and help facilitated by the clan and religion of Bunun resulted in a unique informal care resource [17]. The research findings highlighted that the users of long-term care services harbored a need for spiritual care and religious beliefs significantly help their lives and were a powerful force supporting them in the face of adversities. For Bunun people, a person’s state of health depends on not only the body, but also the force of the “mind” (in the Bunun language called “is-ang”) [28], and the religious force could strengthen the human mind. Under this belief system, tribal churches have naturally become the pivots of religious beliefs in people’s lives. In the long-term care of the Bunun tribe, the churches not only provided material support but also complemented it with mental and spiritual care lacking in public long-term care services. Informal care resources were important and equal to formal care resources in the long-term care system. Owing to the scarcity of formal resources in indigenous tribes, the support of informal resources is indispensable; therefore, when the government implements long-term care policies, it should reinforce and integrate the connection of both [29]. For instance, the possibility of funding and supporting local churches should be considered so that they could perform psychological support services at home. Further, long-term care service development in indigenous villages should not be limited to the fixed model of services in mainstream society but should consider a care model based on the cultural background and local resources. For instance, the government can guide local organizations, such as churches and larger clans in the tribe, with projects to enhance their intention and organize the workforce and material resources for the long-term care service of the tribes. It can empower the tribe members to participate in surveys regarding the people’s demands for long-term care and construct their care model and service operations in order to develop a long-term care service model that meets the needs of tribal culture. These changes will greatly improve the quality of long-term care for indigenous communities in Taiwan.

Lastly, participants generally perceived public services to be beneficial in terms of helping them regain a sense of physical and mental comfort and reducing the burden of their family caregivers; this aligned with the results of a statistical survey on the Taiwanese public’s satisfaction with the overall use of services in the LTC 2.0 [30].

## 5. Conclusions

In terms of accessibility, there is still great room to improve improvement long-term care services for Taiwan’s indigenous tribes. Indigenous people in Taiwan face cultural and economic obstacles to accessing care, such as those related to public transport and human resource development, which are all interconnected. Developing a model for long-term care services tailored to local conditions and rooted in native tribal culture is the most effective strategy for enhancing accessibility, acceptability, and satisfaction with care services in indigenous villages. On this basis, policies formulation must be more respectful, understanding, inclusive, supportive, and flexible.

Based on the results of this study, we would like to make the following suggestions for improving long-term care services for indigenous communities in Taiwan. (1) Local community nurses and long-term care case managers should be empowered, as they are the important bridges between public resources and indigenous villages; specifically, they should be trained and encouraged to actively discover cases and immediately provide information. (2) Efforts should be made to build a local bank of human resources, and recruit and train local indigenous providers of long-term care services so that they serve with cultural sensitivity and with the knowledge of the Bunun language. In order to encourage local indigenous people to join long-term care services, the hours they provide services can be accumulated and when they need services in the future, they can prioritize it at free charge. Moreover, human resource databases and platforms should be established for exchanging care experiences and service innovations. (3) Cultural knowledge, awareness, and competence should be incorporated into the training programs for care personnel working in indigenous tribes. Avoiding prejudice and discrimination is also important. (4) Transportation resource capacity should be checked and complemented; a friendly transportation system should be established; telemedicine and telecare should be combined with preventive medicine and acute care as well as long-term care services in integrated health care centers of indigenous villages.

This study is limited in terms of its generalizability. Owing to a limited workforce and limited materials, the participants of this study included only Bunun long-term care service users in one indigenous village in central Taiwan. To be sure, their experiences might be different from those of other indigenous people in Taiwan; therefore, future research should be expanded to include different tribes.

## Figures and Tables

**Table 1 healthcare-10-02383-t001:** Demographic characteristics of participants (*n* = 14).

Participant No.	Gender	Age	Living Status	Service Items of LTC	Duration of Using Service (Year)
1	Male	43	With spouse only	home services ^1^	10
2	Male	54	Live alone	home services ^1^, transportation services	2
3	Male	57	With spouse only	home services ^1^, professional care service	1
4	Male	61	With spouse only	home services ^1^, transportation services	10
5	Female	73	Live alone	home services ^1^, transportation services	1
6	Male	77	Live alone	home services ^1^, transportation services	3
7	Male	65	With spouse only	home services ^1^, transportation services,	3
8	Male	49	With spouse only	home services ^1^	6
9	Female	78	Live alone	home services ^1^, transportation services, professional care service ^2^	2
10	Male	53	Live alone	home services ^1^	3
11	Female	66	With spouse, children, and grandchild	home services ^1^, transportation services	1
12	Male	50	Live with mother	home services ^1^, transportation services, professional care service ^2^	5
13	Female	65	With children and grandchild	home services ^1^, professional care service ^2^	1
14	Female	71	With children and grandchild	home services ^1^, transportation services	2

Note: ^1^ Home services included personal hygiene, homemaking assistance, food service, and companions to transport to and from medical appointments; ^2^ professional care services included home care and restorative care.

**Table 2 healthcare-10-02383-t002:** An overview of theme, categories, and subcategories.

Theme	Categories	Subcategories
Helpful but still difficult and unfit	Barriers to accessing to long-term care services	Lack of understanding about service information
		Insufficient service resources
		Restrictions on services items
		Economic difficulties
	The need for culturally appropriate long-term care services	Preference for service providers from the same ethnic group
		The desire for mental and spiritual comfort
	Benefits of long-term care services	Physical and mental comfort
		Reduced caregiver burden

## Data Availability

The data presented in this study are available upon request from the corresponding author.

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
