# Peer review of "Experience of Indigenous Peoples’ Access to Long-Term Care Services in Taiwan: A Qualitative Study among Bunun Tribes"

_healthcare, 2022, doi:10.3390/healthcare10122383_

Round 1
Reviewer 1 Report
This is a study describing the care experience of long-term care users in the Bunun tribe. This is an important study to highlight gaps and findings that can be useful to improve healthcare equity.
The following are some minor comments on the reporting for the authors to consider:
1. What were the reasons for stopping at 14 participants? Was data saturation achieved? Were the participants attending one or many of the 16 long-term care stations?
2. It would be helpful to present Tables 1 & 2 where they are cited in-text, instead of placing them at the end of the results section.
3. Were there any limitations in this study? What is the total number of potential participants? Any challenges in recruiting participants?
Reviewer 2 Report
The article deals with the current issue. The research topic is topical globally. The authors described the research methodology well. Characterized the study group in detail.
In my opinion, the article may be published in the Journal.
I have only one comment: references 17 and 28 are the same!!!
Reviewer 3 Report
Dear authors,
Thank you for providing this insight into Taiwanese and indigenous LTC! I find your study interesting, and I have mostly linguistic comments; you should consult someone with good English to improve your language, formulations and concepts.
For example, on l 31 "aging index" The naive reader wonders what this is... Same for "super aged" l 36 On l 49 and 206 you write "offerings" - I believe you mean offers (see dictionary, English is simple but tricky). Genrally you use "the" too much... L 58 "meet the demands"? To demand is different from "need" (cover needs). L 76 interesting about collaboration in farmiing etc., but the savings bank is not active just during the "farming season, right? L 97, 247, 469, 483 "official", you may mean "public"? L 131 "read the text" - not mentioned earlier what this "text" is... l 141 familiar... 147 "revealing" sounds like police work... showing? clarifying?.... 148 "practical care"? Home making? Household chores? Hands-on services? For all the concepts in LTC consult some British or US articles on Home Help etc. to find the adequate concepts normally used. For example: "transportation services" (not "transport service"). Not 412, 422 "body cleaning service", but (help with) "personal hygiene". 199 All LTC clients seem to get the same amount of help (not very much, and not very often), Are there no needs assessments? And please don't repeat "long-term care services" all along. Maybe explain in the beginning, and thereafter just keep to "services" (or, possibly, care services). There are various definitions, but in my view "care" is help with something personal that people can't do themselves. 396 simpler: "blind participant". 421 Table 1 better put it in the beginning, with data and material, design etc. And the lines need better structure, now confusing what belongs to whom.
The same with Table 2 The category "Living arrangement" can be improved: "With spouse only" "with spouse and children" "with brother" etc., not "with family" (the majority are men, and comment on that?). 484 "This is the ultimate---" sounds very demanding... Maybe more humble formulations about improvements, as it seems some clients are at least partly happy with the service(s) they receive? ADDITIONAL >>> 212-216 simplify tribe lives in --- with a long journey --- 288 non-indigenous providers (staff) --- 431 How are authorities spreading information (if at all) about services? Through newspapers? Unclear here... Your suggestions seem OK. 441 Unclear about costs and payment for services... Income-graded? Free for the poorest?
Good Luck with your work!
